# Turning preference in dogs: North attracts while south repels

**Jana Adámková**[ID][1], **Kateřina Benediktová**[1], **Jan Svoboda**[1], **Luděk Bartoš**[2,3], **Lucie Vynikalová**[4], **Petra Nováková**[1], **Vlastimil Hart**[1], **Michael S. Painter**[1], **Hynek Burda**[ID][1]*

**1** Faculty of Forestry and Wood Sciences, Department of Game Management and Wildlife Biology, Czech University of Life Sciences, Praha, Czech Republic, **2** Department of Ethology, Institute of Animal Science, Praha, Czech Republic, **3** Faculty of Agrobiology, Food and Natural Resources, Department of Ethology and Companion Animal Science, University of Life Sciences, Praha, Czech Republic, **4** Faculty of Agrobiology, Food and Natural Resources, Department of Zoology and Fisheries, Czech University of Life Sciences, Praha, Czech Republic

* burda@fld.czu.cz

**Data Availability Statement:** All relevant data are within the paper and its Supporting Information files.

**Funding:** The grant "EVA4.0", No. CZ.02.1.01/0.0/0.0/16_019/0000803 financed by OP RDE to JA,

## Abstract

It was shown earlier that dogs, when selecting between two dishes with snacks placed in front of them, left and right, prefer to turn either clockwise or counterclockwise or randomly in either direction. This preference (or non-preference) is individually consistent in all trials but it is biased in favor of north if they choose between dishes positioned north and east or north and west, a phenomenon denoted as "pull of the north". Here, we replicated these experiments indoors, in magnetic coils, under natural magnetic field and under magnetic field shifted 90˚ clockwise. We demonstrate that "pull of the north" was present also in an environment without any outdoor cues and that the magnetic (and not topographic) north exerted the effect. The detailed analysis shows that the phenomenon involves also "repulsion of the south". The clockwise turning preference in the right-preferring dogs is more pronounced in the S-W combination, while the counterclockwise turning preference in the left-preferring dogs is pronounced in the S-E combination. In this way, south-placed dishes are less frequently chosen than would be expected, while the north-placed dishes are apparently more preferred. Turning preference did not correlate with the motoric paw laterality (Kong test). Given that the choice of a dish is visually guided, we postulate that the turning preference was determined by the dominant eye, so that a dominant right eye resulted in clockwise, and a dominant left eye in counterclockwise turning. Assuming further that magnetoreception in canines is based on the radical-pair mechanism, a "conflict of interests" may be expected, if the dominant eye guides turning away from north, yet the contralateral eye "sees the north", which generally acts attractive, provoking body alignment along the north-south axis.

## Introduction

Dogs in two-choice experiments, when selecting between two dishes with snacks placed in front of them, 90˚ apart, left and right, prefer to turn either clockwise ("right-preferring") or

KB, HB, VH, MSP; the Grant Agency of the Czech University of Life Sciences in Prague, CIGA (Project No. 20174319) to JA, HB, KB, VH, PN, JS, LV; the grant of the Ministry of Agriculture of the Czech Republic MZE-RO0718 to LB.

**Competing interests:** No authors have competing interests.

counterclockwise ("left-preferring") or randomly in either direction ("irresolute"). This turning preference (or non-preference) is individually consistent in all trials but it is biased in favor of north if they choose between dishes positioned north and east or north and west, a phenomenon we denoted as "pull of the north" [1]. This phenomenon was particularly pronounced in older dogs, females, smaller and medium-sized breeds, dogs exhibiting a turning preference, and especially in the north-east choice. We suggested that "pull of the north" represents a further indication of magnetoreception in dogs, the other being non-random directional alignment during marking [2], which was, however, significantly changed when exposed to bar magnets [3], the ability to find a bar magnet [4], or the existence of the so-called "compass run" exhibited during homing [5].

We are, however, aware that for the ultimate evidence of magnetoreception, experiments in defined manipulated magnetic field and/or under conditions of disturbed magnetoreception are necessary. Moreover, the proximate reason for "pull of the north" remains unclear and should be at least hypothesized.

Laterality, i.e. a predictable, non-random preference for using one side of the body (limbs, brain hemisphere, sensory organs) spontaneously or if forced or restricted to choose between two sides, is a known phenomenon in humans and animals. Laterality may be inborn, imprinted, or entrained and has to be taken into account in maze and behavioral two-choice animal experiments [6–10].

Laterality in dogs has been intensively studied with regard to the motoric (efferent) aspect (paw laterality, Kong-test: [11–15]; sensory (afferent) aspect [16–18]; cognitive [19], and emotional aspects [20–22]. Interestingly, and contrary to studies in humans, turning (directional, rotational) preference has remained understudied.

Most people are right-handed, yet tend to instinctively veer to the left upon entering a new space [23]. Interestingly, the counterclockwise action goes also for most athletic tracks, horse and car races, and for baseball players running the bases [24]. There is even evidence that the chariot races at ancient Rome's Circus Maximus ran counterclockwise, too [25,26]. So, in sports, where competitors enter the field of play from the outside of a traced circle, a right-directional choice would lead to a counter-clockwise motion. But when entering the field of action from within the circle—walking out of your apartment to take the dog for a walk, and encountering intersections—right directional choices would tend towards tracing a clockwise path [23]. Interestingly, in the countries, where people drive on the left side of the road, retail shoppers tend to turn counterclockwise—when navigating store aisles, while in the countries, where people drive and keep on sidewalks right, veer clockwise [23]. Tendencies of people to turn either direction are known to architects who use them to design shopping galleries to funnel shoppers in the wished direction [23].

While the preference to turn in a certain direction can be explained by individual inborn laterality (handedness) and experience (facilitation), or–e.g. in the context of our experiment of choice between two dishes, which is a visually guided task, through visual laterality—the "pull of north" is expected to have a magnetoreceptive ground. Examination of this phenomenon has a heuristic potential in getting insight into the very seat and mechanism of magnetoreception, which still remain enigmatic [27].

Sensory laterality (or asymmetry) has been described also in the context of spatial orientation in general and magnetoreception in particular. It has been found that homing pigeons rely more on the right olfactory system in processing the olfactory information needed for the operation of the navigational map [28]. An earlier study [29] has shown that the magnetic compass of a migratory bird, the European robin (*Erithacus rubecula*), was lateralized in favour of the right eye/left brain hemisphere. However, it has been later demonstrated [30] that the described lateralization is not present from the beginning, but develops only as the

birds grow older. In another study [31], it was shown that pigeons can perceive and process magnetic compass directions with the right eye and left brain hemisphere as well as the left eye and right brain hemisphere. However, while the right brain hemisphere tended to confuse the learned direction with its opposite (axial response), the left brain hemisphere specifically preferred the correct direction (angular response). The findings thus demonstrated bilateral processing of magnetic information, but also suggested qualitative differences in how the left and the right brain deal with magnetic cues.

Based on the hitherto knowledge and the above arguments,

1. We hypothesize that if "pull of the north" is due to magnetoreception (and indeed no other explanation is apparent), it should be demonstrated also in an artificial magnetic field shifted by magnetic coils, i.e. the artificially shifted magnetic North should exert the same effect as the natural geomagnetic North.

2. We expect that, consistently with results of the previous study [1] "pull of the north" is more pronounced in "lateralized" dogs and more in the North-East (N-E) combination than in the North-West (N-W) choice.

Furthermore, following questions can be raised (and should be tested) to get insight into the nature of the turning preference:

3. Does the directional preference for turning correlate with motoric laterality (such as paw-laterality, i.e. "handedness")?

4. Is pull of the north a) symmetrical (bilateral, i.e. of the same strength in the clockwise as in counterclockwise direction), or b) asymmetrical (unilateral, i.e. stronger in one particular direction)?

## Material and methods

### Ethics statement

The study did not involve any disturbance or discomfort to the study subjects. The Professional Ethics Commission of the Czech University of Life Sciences in Prague has decided that according to the law and national and international rules, this study has not a character of an animal experiment and does not require a special permit.

### Subjects

Altogether, 23 domestic dogs *Canis familiaris* (11 M, 12 F) from six breeds with pedigree and an average age of 4.8 (± 2.8) years (Table 1) were used in this study. The dogs were pets living in households. All the dog owners were present with their dogs at trials.

### Experimental equipment

The experiment took place in a magnetic coil at the field research station Truba, Kostelec nad Černými lesy, (N 50˚0.40480', E 14˚50.11145') a detached workplace of the Faculty of Forestry and Wood Sciences, Czech University of Life Sciences in Prague, Czech Republic. The magnetic coil (a Merritt coil, built according Kirschvink [32]) was 4 x 4 x 4 m and was located in a separate special building. It was shielded from radiofrequency waves. It was controlled from a separate building next to the coil building. The magnetic field in coils was manipulated by a MagFieldG control software through a GMP4 RJ4.01 control unit and three current amplifiers, each for the Bx axis, the By axis and the Bz axis. The generation system for GMP4 3D coil

**Table 1. List of the tested dogs and resulting indices of directional preference.**

| Dog | Owner | Breed | Sex | Age | Paw motorical laterality | Initial turning preference | Mean turning preference |
|---|---|---|---|---|---|---|---|
| Amalka | KB | Dachshund D | F | 5 | 51 | 17 | 46 |
| Arthur | ES | Dachshund N | M | 2 | n.m. | -32 | -45 |
| Azizi | JS | Beagle | M | 6 | 6 | -4 | 19 |
| Barca | LS | Fox Terrier | F | 12 | 0 | 67 | 42 |
| Bertik | KB | Dachshund D | M | 6 | 8 | -29 | -48 |
| Bessy | JA | Fox Terrier | F | 8 | 26 | 22 | 42 |
| Figy | KB | Dachshund D | F | 5 | 9 | -4 | 8 |
| Gofi | JA | Fox Terrier | F | 3 | -70 | -95 | -96 |
| Hard | JA | Fox Terrier | M | 2 | -46 | 22 | 2 |
| Hugo | KB | Dachshund D | M | 3 | n.m. | -25 | -17 |
| Hurvinek | KB | Dachshund D | M | 7 | 51 | -46 | -45 |
| Jimmy | ES | Dachshund N | M | 2 | n.m. | 60 | 35 |
| Kacka | KB | Dachshund D | F | 5 | 25 | -17 | -18 |
| Kuky | KB | Dachshund D | M | 7 | 27 | 50 | 40 |
| Naty | ES | Münsterländer | M | 3 | n.m. | -45 | -62 |
| Offi | JS | Beagle | F | 9 | -3 | 8 | 6 |
| Pecka | KB | Dachshund D | F | 2 | -44 | 17 | 29 |
| Plysak | KB | Dachshund D | F | 2 | 10 | -37 | -60 |
| Punta | KB | Dachshund D | M | 3 | -1 | -8 | -34 |
| Roxxy | JS | Beagle | F | 9 | 100 | 62 | 67 |
| Shedy | ES | Weimaraner | M | 5 | n.m. | 27 | 40 |
| Sisi | KB | Dachshund D | F | 3 | n.m. | 12 | 42 |
| Zofka | KB | Dachshund D | F | 2 | -20 | 54 | 46 |

Paw motoric laterality = laterality index based on the Kong test; Initial turning preference Turning preference index in the first trials of each dog. Mean turning preference = = mean turning preference index over all trials of each dog. The value of the index can range from -100 to -25 (= left-turning dog) to 25–100 (= right-turning dog). Sex: F = female, M = male. Age is given in years. Dachshund N = normal-sized dachshund, Dachshund D = dwarf-sized dachshund, n.m. = not measured.

system was used to create a defined direct and slowly changing magnetic field and it served to drive the coil system to create a defined magnetic field.

Magnetic induction values in the Cartesian coordinate system (axis $Bx$ = -3225 nT; axis $By$ = 17800 nT; axis $Bz$ = 45448 nT) were set for the experiment, thereby rotating the magnetic field by 90˚ magnetic North was shifted to the topographic (= geomagnetic) East. The magnetic field strength and inclination were maintained as for geomagnetic values for local geographic conditions. The magnetic coil space was used also for the control experiment to test the dogs under local geomagnetic conditions, while other experimental conditions were preserved identic, i.e. shielding of radiofrequency waves, avoiding other influences (wind, sun, outside sounds). The coil room was equipped with cameras (AXIS P5624-E 50HZ—PTZ IP camera, TD / N, 18x zoom, HD 720p, IP66, PoE +) for video recording of the entire experimental space, network speaker with SIP, PoE support (AXIS C3003-E NETWORK HORN SPEAKER, Double—sided audio) and microphone (AXIS T8353A MICROPHONE 3.5MM) at the control station to secure communication of the leading experimenter in the control workplace with two experimenters in the coil.

## Experimental procedure

Dogs were tested indoors, in a room housing the magnetic coils, and should make the choice between two identical dishes. The dishes were placed at a distance of 2.9 m from the point of

release of the dog, always a plus and minus 30˚ from the starting point. Both dishes contained the same treats and dogs were always allowed to empty both. After placing the dishes, the dog was ready for the starting point and waited to obtain a permit to go to a dish. The dogs could not see the placement of the reward dishes. Three experimenters were involved in the experiment; two were present in the magnetic coil (the owner was guarding the dog and prohibited it from seeing the preparation procedure, and the other was preparing the placement of the rewarded dishes), the third experimenter was in the control room using a microphone and headsets to communicate with the two other colleagues, changed the experimental magnetic conditions (switching between control and experimental conditions) according to a randomized schedule and recorded the results (direction of dog first choice) (Fig 1). Note that this person was the only one who knew the actual position of the magnetic North inside the coil.

Each dog was tested in three to five test series under the control conditions with the magnetic North (mN) being 0˚, and in the same number of test series in an artificially shifted magnetic field with mN = 90˚ (where magnetic north was set on topographic east). The order of the test series (control first, shifted field second or shifted field first, control second) was taken into account. Tests series were performed at different days, at different daytimes, evenly distributed over the whole day.

Because a series included four trials in each dish combination alignment (i.e. N-E, E-S, S-W, and W-N), individual dogs experienced either 48 or 80 trials (in 12 or 20 complete series) in which their turning preference (first dish choice) was recorded under control conditions and the same number of records was gathered for experiments in the shifted magnetic field. The difference in the number of series and trials experienced by individual dogs was given by their availability for our study.

In addition, the dog's identity, date, time, sequence of trials combinations, and the order of the trials in the respective series were recorded.

## Paw preferences

To determine paw preference (motoric laterality of dogs), a modified Kong test [e.g. 12,16,33] was used. In this test, it is recorded with which paw (left or right) the dog holds a Kong, a dog toy (KONG Company) when trying to get the food stuffed inside. A plastic yoghurt cup was used instead of Kong. The inner walls and bottom of the cup were covered with a dog's delicacy such as lard, cream cheese. Each dog was tested at home in an open area for 10 minutes while the dog played with the cup and tried to lick it out and the number of touches with either paw was recorded. Simultaneous touches with both paws were also recorded but were not included in the calculation of the index of laterality. The dogs who did not touch the cup during test of paw preference are excluded from the analysis of the Kong test.

## Data analyses

From the recorded choices for each dog, in each trial, the left and right turning preferences were summed, for all four combinations (W-N; N-E; E-S; S-W) separately. For data analysis, the **turning preference index** was calculated in tests performed in the control and shifted magnetic field. The formula (R-L / R + L) x 100 was used, where the R = right and L = left sides are the total numbers of the first choice of left or right dishes. The **laterality index** for the paw preference (Kong test) was calculated using the same formula. The value of the index can range from -100 to -25 (= left-pawed dog) to 25–100 (= right-pawed dog). Dogs with index values between -24 and 24 were considered ambilateral. For the turning preference, altogether ten indices (LI) were calculated; one for each dish combination alignment (N-E, E-S, S-W, and W-N), i.e. four altogether in the control conditions and four altogether in the shifted magnetic

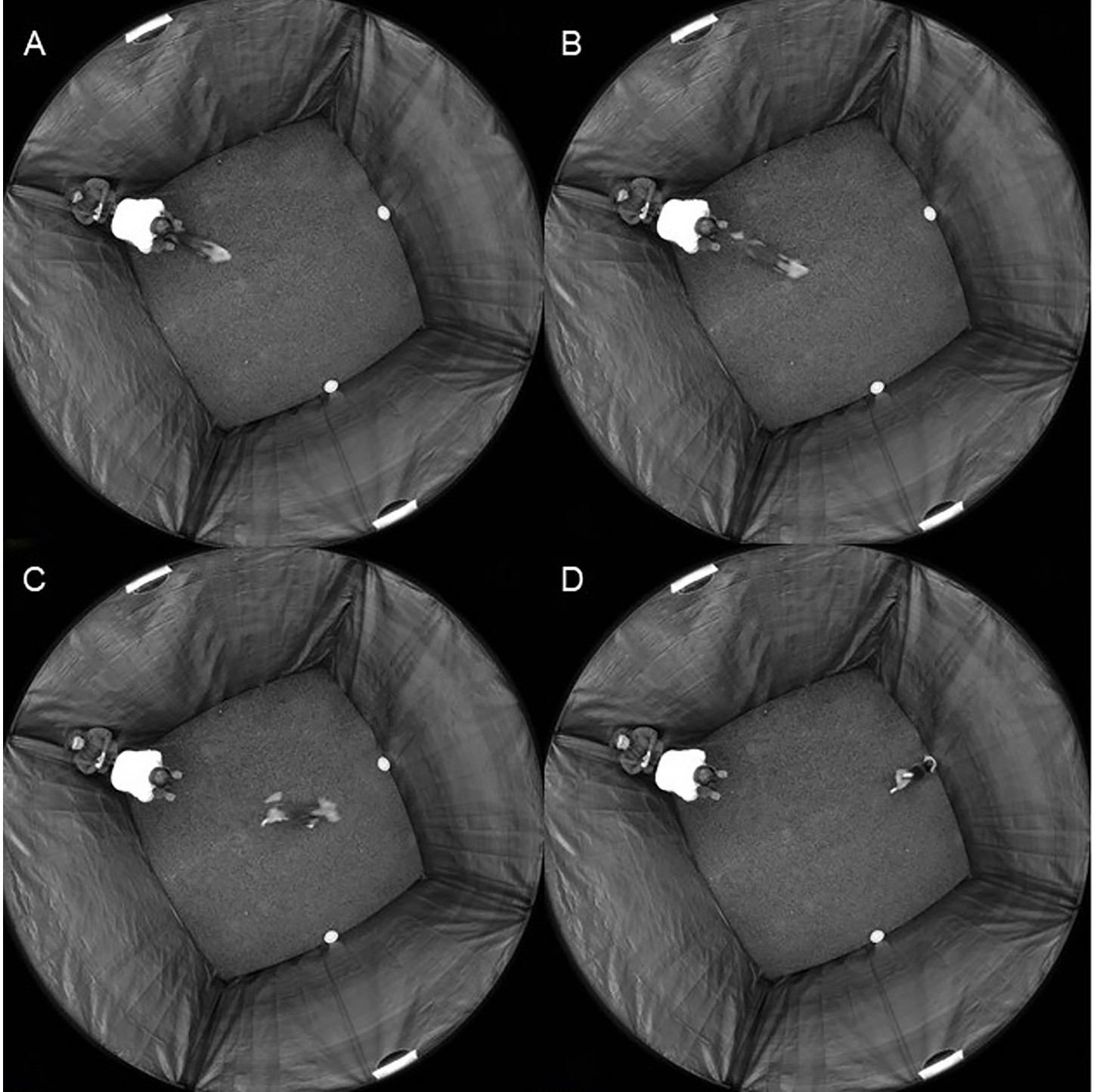

**Fig 1.** Experimental setup as monitored from above by a camera placed at the ceiling of the room, showing the sequence from release of the dog (A) to its choice of one of two dishes (D).

field conditions. Furthermore, we calculated one mean index for control conditions and one mean index for shifted magnetic field (S1 Table). The dogs were divided in turning preference left-preferring, right-preferring or irresolute (ambilateral) preference according to [33] based

on results of the first trials (Initial turning preference in Table 1). Generalized Linear Model (GLM) contained the interaction between Magnetic field and Turning preference classes.

From the recorded choices, preferences for either left or right turn were calculated for all test combinations (N-E, E-S, S-W, W-N) within each trial, and the sum of all trials of each dog. Index of directional preference was then calculated (according to the above formula) for each dog.

All data were analyzed using the SAS System (SAS, version 9.4). For calculating Spearman correlation coefficient we used PROC CORR. To analyze the factors affecting the directional preference index (dependent variable) we used a multivariate Generalized Linear Mixed Model (GLM, PROC MIXED). We constructed two GLMs. The models were applied as a fixed-effect models designed for the repeated measures, i.e., in SAS, with REPEATED = order of testing and the SUBJECT = Name of the dog with compound symmetric covariance structures for repeated measures (TYPE = cs). The first GLM was constructed with the predicted fixed factors Magnetic coil in an interaction with the Turning preference classes, and then we added other variables listed in S2 Table in case they could affect the directional preference index. None of these variables appeared significant and therefore we will not mention them in the text any more. Least squares means (LSMEANs) were calculated for the categorical fixed effects by computing the mean of each treatment and averaging the treatment means. These means of means were then used to compare the factors.

The second model was designed to estimate repeatability of the directional preference across experimental conditions. The GLM contained the only fixed factor Magnetic coil. We calculated repeatability as the intraclass correlation coefficient [34] by adding the RCORR option to the REPEATED.

Independently, mean directional compass preference based on the frequency of first choices in a given combination in all pooled trials was calculated for each dog using circular statistics with Oriana 4.02 (Kovach Computing). Grand mean vectors were then calculated on the base of those mean dog vectors for all the dogs, and subgroups with respect to turning preference, experimental condition, sex, and age.

## Results

### Paw preference (motoric laterality, Kong test)

Following the a priori set criterion, out of altogether 17 dogs tested, 3 dogs were classified as left-lateral, 6 as right-lateral, and 8 as irresolute (ambi-lateral) (Table 1). There was no apparent effect of sex, age, breed or owner on this type of laterality. The correlation between the Kong and overall turning preference tests was rather weak ($r_s$ = 0.317, P = 0.22).

### Turning preference under the control (mN = 0˚) and experimental (mN = 90˚) conditions

Following the a priori set criterion, out of altogether 23 dogs tested, 6 dogs were classified as clockwise-preferring (right-lateral), 7 dogs as counterclockwise-preferring (left-lateral), and 10 as irresolute (ambi-lateral) (Table 1). There was no significant difference in turning preferences of individual dogs between control conditions (mN = 0˚) and the shifted magnetic field conditions (mN = 90˚) (Fig 2). There was a variation in the turning preference index according to the magnetic north direction and Turning preference classes ($F_{(23, 131)}$ = 4.59, P<0.0001, Figs 2 and 3). For the dogs with clockwise turning preference, there was a trend towards increasing the turning preference index from NE, SE, SW and NW. In other words, the clockwise turning dogs exhibited the lowest turning preference index in the combination North-

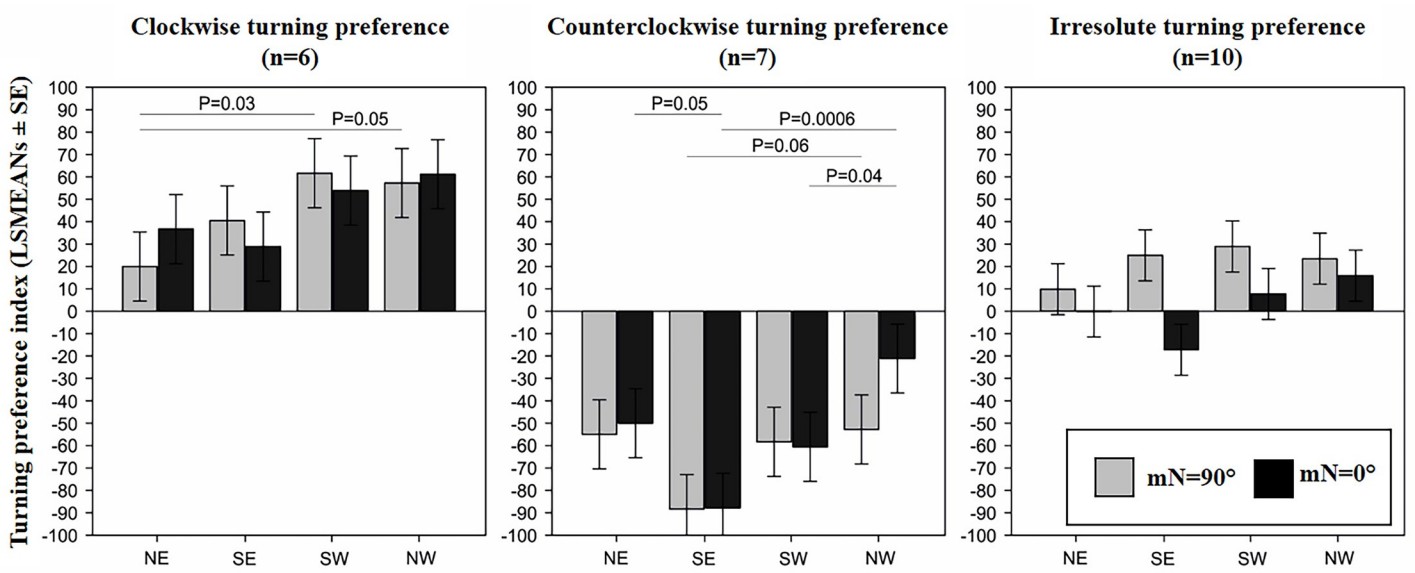

**Fig 2. Turning preference index.** (Least Square Means ± SE) for clockwise-preferring (left), counterclockwise-preferring (middle), and irresolute (right) dogs under the conditions of the magnetic North (mN) = 0˚ (control) and mN = 90˚ (shifted magnetic field) for the four particular combinations of the placement of dishes.

East. However, only the difference between NE vs NW and between NE and SW, and only in the shifted magnetic field, reached the level of significance (P = 0.05) (Fig 2 left). For the dogs with counterclockwise turning preference, the most intensive counterclockwise preference was shown in SE orientation in comparison with NW and partly NE, while the weakest preference was in shown in the NW combination. Significant differences were achieved in the shifted magnetic field in SE vs NW, and under control conditions in NE vs SE, SE vs NW, SW vs NW (Fig 2, middle). No trend nor differences were detected for dogs showing irresolute turning preference (Fig 2, right).

There was significant bias from the overall turning preferences in the eastern hemisphere, expressed as the "pull of the north", in that a dish placed eastwards was more frequently chosen than a dish placed southwards and a dish placed northwards more frequently chosen than a dish placed eastwards, resulting in an average (theoretical) preference for NNE (Fig 4, Table 2). In a more differentiated view, this result was due to a dominant preference of females and/or clockwise preferring dogs for North (over East) and to an additional weaker pull of the East over South in males and/or counterclockwise preferring dogs. "Pull of the north" in irresolute dogs was indicated but not significant (Table 2, Figs 2 and 3).

**Repeatability of turning preference.** A single factor of Magnetic coil was not significant ($F_{1, 22} = 1.16$, P = 0.86). On the other hand, Repeatability was high (r = 0.76).

## Discussion

Turning preference did not correlate with the motoric paw laterality (Kong test). Apparently, both types of preferences are controlled by different proximate mechanisms / pathways. This conclusion is consistent with earlier findings [35] showing that visual (sensory) and paw (motoric) laterality in dogs are independent of each other. None of the dogs had any previous experience with emptying cups (i.e. Kong-type tests). None of the dogs used in this study had a history of being trained "Heel" to come and follow the master at her/his left (or right) side. Consequently, their turning preferences can be considered natural, spontaneous, inborne, and

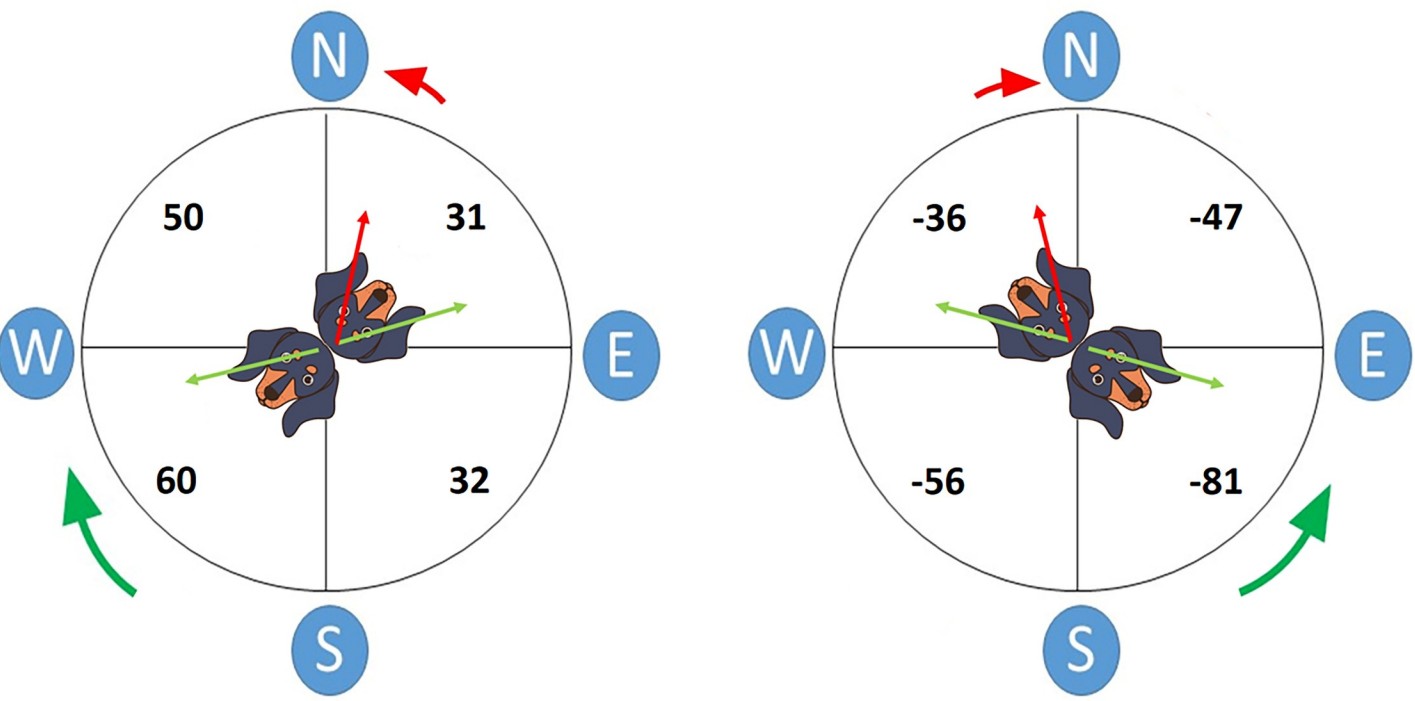

**Fig 3. Numbers in each quadrant (in the respective four compass combinations: N-E, E-S, S-W, W-N) show mean values of turning preference indices calculated from individual dogs and pooled across all trials (both control and shifted magnetic field conditions).** The value of the index can range from -100 to -25 (= left-turning dog) to 25–100 (= right-turning dog). Data were partitioned by turning preference (left figure shows clockwise turning preference, right figure shows counterclockwise turning preference; irresolute dogs were not calculated. The green arrow over the dog's head in the centre of the circle indicates the direction of view of the (supposedly) dominant eye which guides turning direction, while the red arrow shows the direction of view of the contralateral eye, supposed to exert "pull of the north" if heading northwards. Green arrow outside the circle designates the preferred direction of turning, the shorter red arrow designates "pull of the north".

not entrained. Accordingly, there was no significant difference in the turning preference in particular dogs between the first and second experimental series and there was no effect of the respective owner. Interestingly, among the dogs who turned clockwise there were more females, while among the dogs turning counterclockwise there were more males. The sample was, however, too small to allow any general conclusion with regard to the effect of sex on turning preference. In fact, no clear effect of sex on turning preference was found in a previous study (with a different composition of the study sample) [1].

Consistently with results, of the previous study in open field [1], the turning preference was consistent for each particular dog for all combinations of placement of dishes also in an interior with uniform walls, no apparent landmarks, and no sun or wind cues. Concordantly with the results of the previous study, this preference was slightly, yet significantly disturbed (or pronounced) in that the north-placed dishes were more frequently chosen than would be expected according to the average turning preference of each particular dog. Most important in the context of the present study is the finding that, magnetic and not topographic, north affected the mentioned bias.

The detailed analysis shows, however, that the "pull of the north" is a more complex phenomenon, involving also "repulsion of the south". These effects are unilateral: the clockwise turning preference in the right-preferring dogs is more pronounced ("accelerated") in the S-W combination, while the counterclockwise turning preference in the left-preferring dogs is

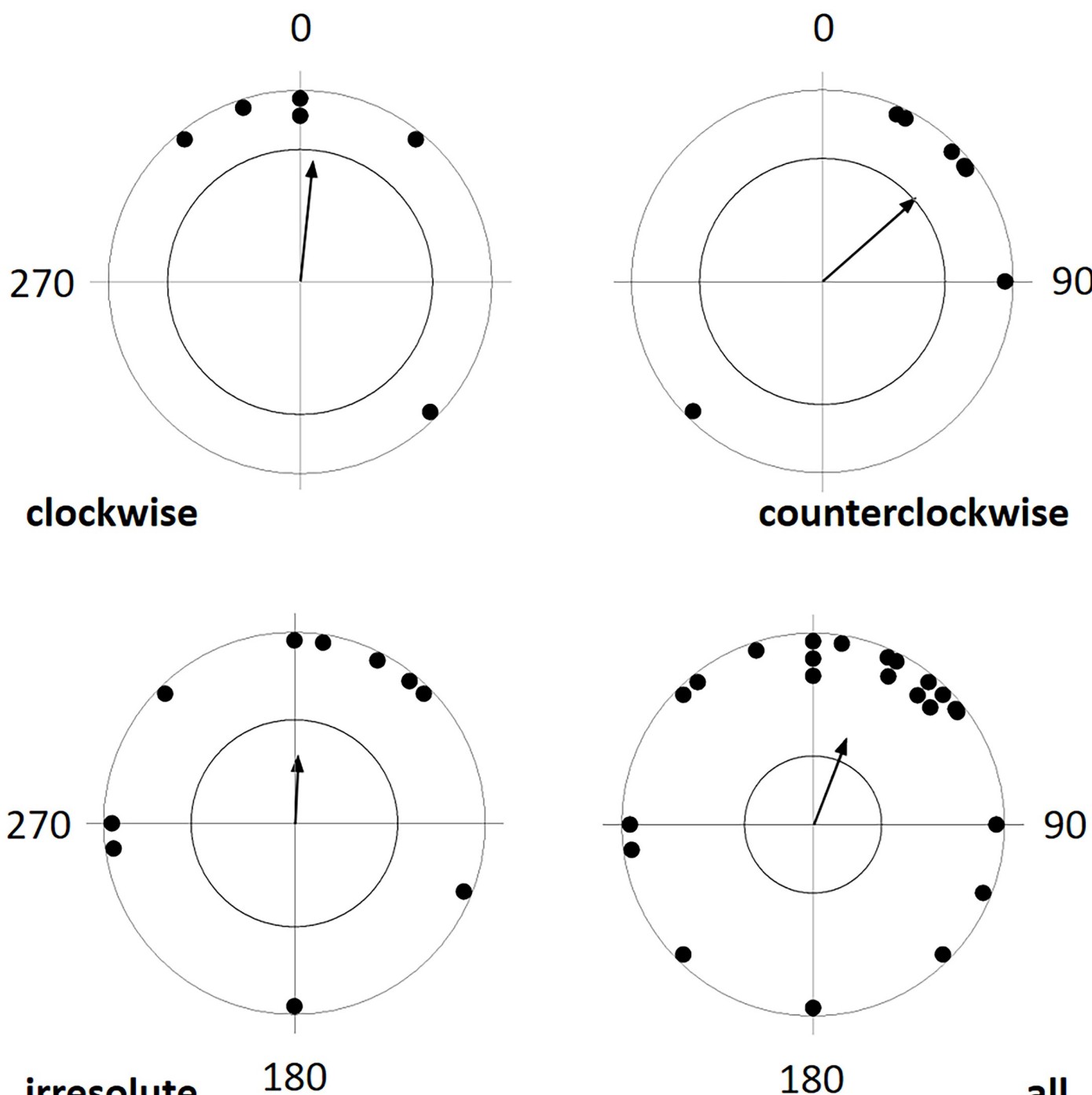

**Fig 4. Mean preference for compass direction of a dish with snacks of the first choice.** Angular means over dogs preferring to turn clockwise, those preferring to turn counterclockwise, dogs which were irresolute in their preference, and over all dogs. The arrow indicates the grand mean axial vector (μ) calculated over all angular means. The length of the mean vector (r) provides a measure of the degree of clustering in the distribution of the mean vectors. The inner circle marks the 0.05 level of significance border of the Rayleigh test. See Table 2 for statistics.

"accelerated" in the S-E combination. On the other hand, N-E combination decreases ("decelerates") clockwise turning preference in the right-preferring dogs, while in the N-W combination, the counterclockwise turning preference in the left-preferring dogs will be reduced. In

**Table 2. Circular statistics for frequencies of choices of a dish placed in different cardinal compass directions in front of a dog in dual choice experiments, where the dog chose between north or east, east or south, south or west, west or north.**

| Variable | All trials | mN = 0˚ | mN = 90˚ | 1st series | 2nd series |
|---|---|---|---|---|---|
| Number of dogs tested | 23 | 23 | 23 | 23 | 23 |
| Mean vector (µ) | 21˚ | 43˚ | 350˚ | 22˚ | 17˚ |
| Length of mean vector (r) | 0.485 | 0.557 | 0.464 | 0.347 | 0.566 |
| Circular standard deviation | 69˚ | 622˚ | 71˚ | 83˚ | 61˚ |
| 95% Confidence interval (-/+) for µ | 349˚-53˚ | 16˚-70˚ | 316˚-23˚ | 335˚-68˚ | 351˚-44˚ |
| 99% Confidence interval (-/+) for µ | 339˚-63˚ | 7˚-78˚ | 305˚-34˚ | 321˚-82˚ | 342˚-52˚ |
| Rayleigh test (Z) | 5.402 | 7.134 | 4.945 | 2.766 | 7.378 |
| Rayleigh test (p) | 0.004 | 4.92E-04 | 0.006 | 0.061 | 3.68E-04 |
| Variable | males | females | clockwise preferring | counterclockwise preferring | irresolute |
| Number of dogs tested | 11 | 12 | 6 | 7 | 10 |
| Mean vector (µ) | 44˚ | 0˚ | 6˚ | 48˚ | 3˚ |
| Length of mean vector (r) | 0.517 | 0.529 | 0.637 | 0.655 | 0.356 |
| Circular standard deviation | 66˚ | 65˚ | 54˚ | 53˚ | 82˚ |
| 95% Confidence interval (-/+) for µ | 358˚-90˚ | 318˚-42˚ | 318˚-53˚ | 9˚-90˚ | 279˚-87˚ |
| 99% Confidence interval (-/+) for µ | 344˚-105˚ | 304˚-56˚ | 303˚-68˚ | 354˚-103˚ | 252˚-113˚ |
| Rayleigh test (Z) | 2.939 | 3.357 | 2.431 | 3.005 | 1.268 |
| Rayleigh test (p) | 0.049 | 0.031 | 0.084 | 0.043 | 0.288 |

Each compass direction was offered with the same frequency. Mean vectors in this table represent thus grand mean vectors. Cf. Fig 4.

this way, in the total, south-placed dishes are less frequently chosen than would be expected, while the north-placed dishes are apparently more preferred. Since "rotational deceleration" is stronger in N-E than the N-W combination, while the "acceleration" is stronger in the S-E than in the S-W combination, the resulting theoretical mean preference is for Northeast.

It may be of relevance and significance in this context that the analysis of published results on magnetic alignment behaviour in a variety of vertebrate species revealed that magnetic alignment typically coincides with the north-south magnetic axis, however, the mean directional preferences of an individual or group of organisms is often rotated clockwise from the north-south axis [36–38]. The deviation from the magnetic north-south axis could originate at different levels in the sensory hierarchy: it could be related either to asymmetries at the sensor level or to functional brain asymmetries, i.e. central processing.

Although the mode of the perception of the magnetic compass direction in animals remains enigmatic [27], findings from behavioral, histological, neuroanatomical, and electrophysiological studies have led to several physically viable theoretical models that might also apply to dogs. Two mechanisms are most widely discussed in the literature: the magnetite-based mechanism and the radical-pair mechanism.

Perhaps the most intuitively appealing mechanism to explain magnetosensitivity in animals is the idea of a small permanent magnet inside the animal that acts like a compass needle [39]. Magnetite-based sensors may be located anywhere in the body, they do not need to be concentrated in (paired) organs and they can be very tiny.

Another proposed mechanism for magnetoreception in animals is based on an effect of the magnetic field on the quantum spin states of a photo-excited chemical reaction that forms long-lived, spin correlated radical pair intermediates (radical pair mechanism; [40,41]. It is believed to occur in the specialized retinal cells [42,43]. It is assumed that the magnetic field may generate a "visual" pattern of varying light intensity, color, and/or contrast superimposed on the normal visual scene [40,44,45]. The model suggests that north or south "patterns" are

more clearly recognizable and easier to be followed than east or west "patterns". Accordingly, and alternatively, the "pull of the north" could be also interpreted as a "deflection / repellence by the east or west".

Given that the choice of a dish in our experiment was visually guided, we may postulate that the turning preference was determined by the dominant eye, so that a dominant right eye resulted in clockwise, and a dominant left eye in counterclockwise turning. Assuming further that magnetoreception in canines is based on the radical-pair mechanism [46,47], a "conflict of interests" may be expected, if the dominant eye guides turning away from north, yet the contralateral eye "sees the north", which generally acts attractive, provoking body alignment along the north-south axis. To test this hypothesis, visual dominance (eyedness) in particular dogs should be studied in an independent test, e.g. sensory jump test [35].

Magnetic alignment might have an adaptive function in that it provides a global reference frame that helps to structure and organize spatial behavior and perception over many different spatial scales. For example, one possibility is that magnetic alignment helps to put the animal into register with a known orientation of a mental (cognitive) map, reducing the complexity of local and long-distance navigation, and reduces the demands on spatial memory [44]. This would be analogous to strategies used in human orientation; it is much simpler and intuitive to navigate when the navigators align themselves with a physical map (i.e. the users rotate their body direction to coincide with the alignment of the physical map), rather than to navigate by mentally rotating the physical map to align with the user's orientation. Therefore, we suggested that the mental map in animals is fixed in alignment with respect to the magnetic field [2,38]. Indeed, important component(s) of the cognitive map may be derived from the magnetic field (see below) and spontaneous magnetic alignment behavior may help to place the animal into register with this map. This relatively simple alignment strategy would help animals to reliably and accurately 'read' their cognitive map and/or extend the range of their maps when exploring unfamiliar environments. Accordingly, animals of different taxa were frequently reported to prefer to head about northwards when feeding (reviewed in [36–38]).

We suggest that the described simple turning test has a high heuristic potential and should be extended for tests of visual laterality and be performed under a wider array of experimental conditions to get more insight into the very mechanism, seat and function of magnetoreception.

## Supporting information

**S1 Table. Indices of laterality each tested dog.** ITP = index of turning preference; W-N designates the combination in which the test dishes were placed west and north; N-E = designates the combination in which the test dishes were placed north and east; E-S designates the combination in which the test dishes were placed east and south; S-W = designates the combination in which the test dishes were placed south and west, mN = 0˚ designates a control experiment; mN = 90˚ designates an experimental condition with a shifted magnet field.
(DOCX)

**S2 Table. Variables available for statistical analysis.**
(DOCX)

**S3 Table. Source data.**
(XLSX)

## Author Contributions

**Conceptualization:** Jana Adámková, Kateřina Benediktová, Vlastimil Hart, Hynek Burda.

**Data curation:** Jana Adámková, Kateřina Benediktová.

**Formal analysis:** Jana Adámková, Kateřina Benediktová, Luděk Bartoš, Hynek Burda.

**Funding acquisition:** Jana Adámková, Kateřina Benediktová, Jan Svoboda, Luděk Bartoš, Vlastimil Hart, Hynek Burda.

**Investigation:** Jana Adámková, Kateřina Benediktová, Jan Svoboda, Lucie Vynikalová, Petra Nováková, Michael S. Painter.

**Methodology:** Jana Adámková, Kateřina Benediktová, Vlastimil Hart, Michael S. Painter, Hynek Burda.

**Project administration:** Jana Adámková, Kateřina Benediktová, Vlastimil Hart, Hynek Burda.

**Resources:** Vlastimil Hart, Hynek Burda.

**Supervision:** Jana Adámková, Kateřina Benediktová, Vlastimil Hart, Hynek Burda.

**Validation:** Jana Adámková, Kateřina Benediktová, Hynek Burda.

**Visualization:** Jana Adámková, Luděk Bartoš, Hynek Burda.

**Writing – original draft:** Jana Adámková, Kateřina Benediktová, Luděk Bartoš, Michael S. Painter, Hynek Burda.

**Writing – review & editing:** Hynek Burda.

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
