## [Decision Letter · Decision Letter 0]

20 Oct 2020

PONE-D-20-31110

Turning preference in dogs: north attracts while south repels

PLOS ONE

Dear Dr. Burda,

Thank you for submitting your manuscript to PLOS ONE. After careful consideration, we feel that it has merit but does not fully meet PLOS ONE’s publication criteria as it currently stands. Therefore, we invite you to submit a revised version of the manuscript that addresses the points raised during the review process.

The reviewers believed there was merit to the study, but also that they lacked all the information necessary to make firm conclusions on the result's validity and the conclusions drawn.  Specifically, both reviewers felt the methods sections was not sufficiently clear.  I believe that much more attention to the methods will be needed for publications.   Specifically the clarification of abbreviations, and the number of trials and what where the indices were used.  I felt that the description of how the experimenters were blinded also needed a great deal more clarity - e.g., state very concisely who interacted with the dogs, and who new the location of magnetic north in each experiment.  

The results appeared to leave the reviewers with several more questions.  It is unfortunate that Fig4 was missing for the reviewers, however, both reviewer #1 and #2 had additional important questions on the interpretations of figs 2 and 3.  Although new experiments may not be needed, Reviewer #1 had some important questions regarding dog-owner interactions that should be addressable and would hopefully help eliminate some potentially trivial explanations for the North pull.   Additional questions on eye-laterality were brought up in response to the discussion.  I believe these are important questions from reviewer #1 and please try to answer if possible. If you cannot answer them, these limitations of current knowledge should be at least addressed in the discussion. 

There were also questions regarding the role of the eyes in magnetic detection.  Unless you have experimental data that addresses this, I believe this can also be handled in the discussion by referring to data from other vertebrate systems and clearly drawing the inferences.  

We look forward to receiving your revised manuscript.

Kind regards,

Gregg Roman, PhD

Academic Editor

PLOS ONE

Journal Requirements:

2. Please upload a copy of Figure 4, to which you refer in your text on page 10. If the figure is no longer to be included as part of the submission please remove all reference to it within the text.

3. We note that Figure 1 includes an image of participants in the study. 

As per the PLOS ONE policy (http://journals.plos.org/plosone/s/submission-guidelines#loc-human-subjects-research) on papers that include identifying, or potentially identifying, information, the individual(s) or parent(s)/guardian(s) must be informed of the terms of the PLOS open-access (CC-BY) license and provide specific permission for publication of these details under the terms of this license.

Please download the Consent Form for Publication in a PLOS Journal (http://journals.plos.org/plosone/s/file?id=8ce6/plos-consent-form-english.pdf). The signed consent form should not be submitted with the manuscript, but should be securely filed in the individual's case notes.

Please amend the methods section and ethics statement of the manuscript to explicitly state that the patient/participant has provided consent for publication: “The individual in this manuscript has given written informed consent (as outlined in PLOS consent form) to publish these case details”.

If you are unable to obtain consent from the subjects of the photograph, you will need to remove the figure and any other textual identifying information or case descriptions for these individuals.

Reviewers' comments:

Reviewer's Responses to Questions

**Comments to the Author**

1. Is the manuscript technically sound, and do the data support the conclusions?

Reviewer #1: Partly

Reviewer #2: Yes

2. Has the statistical analysis been performed appropriately and rigorously? 

Reviewer #1: Yes

Reviewer #2: I Don't Know

3. Have the authors made all data underlying the findings in their manuscript fully available?

Reviewer #1: Yes

Reviewer #2: Yes

4. Is the manuscript presented in an intelligible fashion and written in standard English?

Reviewer #1: Yes

Reviewer #2: No

5. Review Comments to the Author

Reviewer #1: There were a few places where I was a little confused about the method. For example the paragraph beginning on Line 193. I was confused about how many total trials dogs had and what exactly what meant by first choice. The sentence on Lines 220-222 I found similarly confusing. What were the 10 indices? There were not 10 mentioned so I wasn’t sure what was being explored here.

Do other studies show similar patterns in laterality? That is, that about half of the dogs do not have a preference? Is laterality preference (or lack thereof) independent of the task? Or could a dog have a preference for one paw on a kong task and a preference for another paw on a different task? It could be helpful to show that the dogs’ paw laterality isn’t just random but is stable within dogs either by citing work that demonstrates that it is so or by giving these dogs a second laterality test and showing that they are consistent.

I think the talk of eye dominance is interesting, but the authors’ case would be significantly strengthened by demonstrating that such eye preferences exist. In other words, is there a simple eye dominance task that the authors could do to assess the dogs’ eye dominance? Given that so much of their discussion is based on the assumption about eye dominance it would strengthen the paper significantly to show that such eye dominance exists and tracks with their predictions.

Similarly, is there evidence that dogs have magnetic field receptors in their eyes?

The authors mention that the dogs do not have a history of coming to heel, but I wonder about other types of owner interactions. Do owner handedness or owner turning preferences track with dogs’ preferences?

There are a few awkward sentences (lines 85-88; 132-134)

Where is Figure 4?

Reviewer #2: In earlier outdoor experiments, dogs were found to prefer the North direction and avoid South, when choosing between two food dishes placed in front of them. In this paper, the authors repeat these experiments indoors under controlled conditions, and by shifting the north direction of the magnetic field, they demonstrate that this preference is based on the magnetic field.

In the introduction, the questions are clearly stated. The description of the experimental procedure appears rather cryptic and suffers from the use of many abbreviations. The result part is hard to read. I have problems to derive from Fig. 2 a “pull of the north”, and also in Fig. 3, it is unclear what the numbers in each quadrant mean. Fig. 4 is missing altogether.

The best part of the paper is definitely the discussion. I welcome the authors’ attempt to propose a promising idea for explaining why so many animals show a magnetic alignment.

6. PLOS authors have the option to publish the peer review history of their article (what does this mean?). If published, this will include your full peer review and any attached files.

Reviewer #1: No

Reviewer #2: No

---

## [Author Response · Author response to Decision Letter 0]

7 Dec 2020

Dear editor, dear reviewers,

we appreciate very much your interest in our study and attention given to our manuscript and the very thoughtful and constructive comments aimed to improve our contribution. We have carefully considered all of them, and we response to them point by point and describe how we changed the manuscript (below, in blue print). Thank you very much for giving us the chance to revise and amend the ms and for taking the revised version and our rebuttal into consideration.

Best regards

Hynek Burda

On behalf of all coauthors

Comments and summary of the academic editor

Editor points out the need for:

clarification of abbreviations, and the number of trials and what where the indices were used. 

This clarification was requested also by the reviewers and we respond specifically below. Also, we have addressed these points in the revised ms.

the description of how the experimenters were blinded (e.g., state very concisely who interacted with the dogs, and who new the location of magnetic north in each experiment). 

We describe this on lines 177-185 of the revised ms. We added further information which points out the way of blinding the experiment.

Fig4 was missing 

This is indeed unfortunate and we apologize. The figure has been uploaded now. Please note that the statistical values in Table 2 have partly changed. This is because we based the figure now on the attribution of dogs to laterality categories according to results in the first trials (and not to mean laterality indices) to ensure comparability of all figures.

both reviewer #1 and #2 had additional important questions on the interpretations of figs 2 and 3.

We react below and in the revised ms.

Although new experiments may not be needed, Reviewer #1 had some important questions regarding dog-owner interactions that should be addressable and would hopefully help eliminate some potentially trivial explanations for the North pull. Additional questions on eye-laterality were brought up in response to the discussion. 

We address these important and interesting points below and in the revised ms.

Reviewer #1: 

There were a few places where I was a little confused about the method. For example the paragraph beginning on Line 193. I was confused about how many total trials dogs had and what exactly what meant by first choice.

Looking at the sentence with time lag, we see also the problem and agree with the reviewer. We have reworded the sentence and specified the numbers as follows:

Because a series included four trials in each dish combination alignment (i.e. N-E, E-S, S-W and W-N), individual dogs experienced either 48 or 80 trials (in 12 or 24 complete series) in which their turning preference (first dish choice) was recorded under control conditions and the same number of records was gathered for experiments in the shifted magnetic field. The difference in the number of series and trials experienced by individual dogs was given by their availability for our study.

The sentence on Lines 220-222 I found similarly confusing. What were the 10 indices? There were not 10 mentioned so I wasn’t sure what was being explored here.

Again, we agree with the reviewer and apologize. We have changed the text as follows:

For the turning preference, altogether ten indices (LI) were calculated; one for each dish combination alignment (N-E, E-S, S-W, and W-N), i.e. four altogether in the control conditions and four altogether in the shifted magnetic field conditions. Furthermore, we calculated one mean index for control conditions and one mean index for shifted magnetic field.

Do other studies show similar patterns in laterality? That is, that about half of the dogs do not have a preference? Is laterality preference (or lack thereof) independent of the task? Or could a dog have a preference for one paw on a Kong task and a preference for another paw on a different task? It could be helpful to show that the dogs’ paw laterality isn’t just random but is stable within dogs either by citing work that demonstrates that it is so or by giving these dogs a second laterality test and showing that they are consistent.

Concerning general laterality behavior in dogs: These are relevant questions, but it is not really what this study set out to address. There are published papers on laterality in dogs on a variety of behavioral tasks which we mention and cite (reference numbers 6-22), however, what we can say is that laterality, as measured by the Kong test, cannot account for the turning preference exhibited by the dogs evaluated. Indeed, this is a strength of the study rather than a weakness, as it uncouples handedness/lateral dominance (motoric laterality) and magnetic turning preference (presumably a sensory laterality). Similarly, it was shown in an earlier report (Tomkins et al. 2010, a newly added reference) that visual (sensory) and paw (motoric) laterality are independent of each other (see also the point below).

I think the talk of eye dominance is interesting, but the authors’ case would be significantly strengthened by demonstrating that such eye preferences exist. In other words, is there a simple eye dominance task that the authors could do to assess the dogs’ eye dominance? Given that so much of their discussion is based on the assumption about eye dominance it would strengthen the paper significantly to show that such eye dominance exists and tracks with their predictions.

Unfortunately, at the current stage of research, given the (wo)manpower and current lockdown-like restrictions we're not going to carry out an additional eye dominance experiment to satisfy this request. For sure, and as we state in the text, it is an inspiration and suggestion for follow-up research. We can, nevertheless, bolster the argument of eye dominance with previous studies from vertebrates, with a focus on mammals that show eye dominance/laterality plays a large role in behavioral ecology. There's lots of examples from birds (where different eyes can specialize on different tasks – either for foraging or for surveillance; the eye dominance with reference to magnetoreception in birds was reported in Refs.29-30, cited in our manuscript). Moreover, eye dominance = ocular dominance = eye preference = eyedness is a well known phenomenon to human ophthalmologists (see Wikipedia and basic literature cited there) and there is no reason to assume that dogs would be different from humans in this respect. Indeed, there is one paper published explicitly on this topic, which, unfortunately, was not cited in the first version of the ms, but, fortunately, came to our notice now to be cited in the revised version (Ref. 35 in the revised version).

Similarly, is there evidence that dogs have magnetic field receptors in their eyes?

There is no direct evidence, but it has been suggested multiple times for canines in previous published studies (Refs. 45-46) and it has been discussed as the putative magnetoreception mechanism in terrestrial vertebrates (Refs. 39-44), with subterranean mammals who evolved under completely different environmental/ecological contexts, being the exception (Ref. 37). Given the robust support for a photoreceptor based mechanism in closely related taxa, it is an interesting hypothesis to propose and is justified based on previous findings from a diverse array of vertebrates, including mammals. 

The authors mention that the dogs do not have a history of coming to heel, but I wonder about other types of owner interactions. Do owner handedness or owner turning preferences track with dogs’ preferences?

Unrelated to the study and the use of different magnetic field alignments clearly shows that the pull of magnetic north mediates these behaviors. It might be interesting look at some of these other factors, however the study design was intended to address the questions outlined in the last paragraph of the introduction. 

There are a few awkward sentences (lines 85-88; 132-134)

In fact the sentence on lines 85-88 is a word-by-word citation from an English book (Ref. 23). 

We have shortened it and slightly reworded it now and hope that it became more straightforward.

Also the second criticized sentence was reworded.

Where is Figure 4?

As admitted above – this was an unfortunate omission and the figure 4 has been uploaded now.

Reviewer #2: In earlier outdoor experiments, dogs were found to prefer the North direction and avoid South, when choosing between two food dishes placed in front of them. In this paper, the authors repeat these experiments indoors under controlled conditions, and by shifting the north direction of the magnetic field, they demonstrate that this preference is based on the magnetic field.

In the introduction, the questions are clearly stated. 

We are pleased by this assessment.

The description of the experimental procedure appears rather cryptic and suffers from the use of many abbreviations.

The reviewer might be right but the problem can be solved only on costs of losing some details or lengthening the text and making it even less understandable. Importantly, all abbreviations are either known units (nT = nanotesla), or are commonly used in the literature of this kind and in any case explained when first used (M = male, F = female, N = North, magN = magnetic North, etc.), or explained when used in a formula (R = right, L = left) or they specify marks of the used software or hardware (which information is important for those who would like to assess suitability of our equipment or replicate the experiments and equip their labs. Finally, there are, abbreviations which are of importance and interest only for statisticians who themselves are familiar with statistical Analysis system (SAS), e.g. LSM for Least squares means, GLM (generalized linear model), etc. Again, all these abbreviations are explained when first mentioned. Repeating whole descriptions in each sentence instead of using these abbreviations would not make the text more fluent, readable and understandable. Omitting these and not mentioning these models would, for sure, be criticized by statisticians.

The result part is hard to read. 

We reworded and complemented the text.

I have problems to derive from Fig. 2 a “pull of the north”

The reviewer is very attentive. There was a mistake in the Figure 2a. The corresponding author uploaded mistakenly an earlier and incorrect version of the figure. Sorry for that error and thank you for finding it. The correct version is uploaded now and the figure is explained in more detail.

and also in Fig. 3, it is unclear what the numbers in each quadrant mean. 

Description of the Fig. 3 is reworded as follows:

Fig 3: Numbers in each quadrant (in the respective four compass combinations (N-E, E-S, S-W, W-N) show mean values of turning preference calculated from individual dogs and pooled across all trials (both control and shifted magnetic field alignments). Data were partitioned by turning preference (left figure shows clockwise turning preference, right figure shows counterclockwise turning preference; irresolute dogs were not calculated. The green arrow over the dog's head in the…….. (the further description of the figure remains unchanged).

Fig. 4 is missing altogether.

As admitted above – we apologize for this unfortunate omission. The figure 4 has been uploaded now.

The best part of the paper is definitely the discussion. I welcome the authors’ attempt to propose a promising idea for explaining why so many animals show a magnetic alignment.

We appreciate this opinion.

---

## [Decision Letter · Decision Letter 1]

5 Jan 2021

PONE-D-20-31110R1

Turning preference in dogs: north attracts while south repels

PLOS ONE

Dear Dr. Burda,

Thank you for submitting your manuscript to PLOS ONE. After careful consideration, we feel that it has merit but does not fully meet PLOS ONE’s publication criteria as it currently stands. Therefore, we invite you to submit a revised version of the manuscript that addresses the points raised during the review process.

Along with the two reviewers, I believe your manuscript is very nearly ready for publication. I appreciate the detail with which you address the reviewers comments and concerns.  Reviewer #1 had some last edits that you may wish to consider.  Please consider these edits before returning the manuscript.  I would especially like you to consider a change to the Fig1 legend that would make the meaning of the measures more apparent to someone less familiar with the experimental approach.   Please also read through closely to make sure you catch any other existing typos before resubmission. 

Congratulations on a very nice paper.       

We look forward to receiving your revised manuscript.

Kind regards,

Gregg Roman, PhD

Academic Editor

PLOS ONE

Reviewers' comments:

Reviewer's Responses to Questions

**Comments to the Author**

1. If the authors have adequately addressed your comments raised in a previous round of review and you feel that this manuscript is now acceptable for publication, you may indicate that here to bypass the “Comments to the Author” section, enter your conflict of interest statement in the “Confidential to Editor” section, and submit your "Accept" recommendation.

Reviewer #1: (No Response)

Reviewer #2: All comments have been addressed

2. Is the manuscript technically sound, and do the data support the conclusions?

Reviewer #1: Yes

Reviewer #2: Yes

3. Has the statistical analysis been performed appropriately and rigorously? 

Reviewer #1: I Don't Know

Reviewer #2: Yes

4. Have the authors made all data underlying the findings in their manuscript fully available?

Reviewer #1: Yes

Reviewer #2: Yes

5. Is the manuscript presented in an intelligible fashion and written in standard English?

Reviewer #1: Yes

Reviewer #2: Yes

6. Review Comments to the Author

Reviewer #1: Some minor things:

Lines 153-162 — paragraph should be in past tense

Line 219 tense shift

Line 244 “according to based on” (redundant)

Line 326 “more frequently" need to add "chosen”

In the figure caption for Fig 1 please give a sense of what these numbers mean. - ie - left and + right? We don't need the full explanation that comes later in the text, but a sense of what these numbers are telling us would be helpful at this point.

I still find the figures confusing, but I’m not an expert on how to present these results with turning preference so I leave it to the other reviewer and the editor's expertise.

Reviewer #2: The text is considerably improved, and it is now easier to follow the argumentation of the authors. I think that the paper can be published now.

7. PLOS authors have the option to publish the peer review history of their article (what does this mean?). If published, this will include your full peer review and any attached files.

Reviewer #1: No

Reviewer #2: No

---

## [Author Response · Author response to Decision Letter 1]

7 Jan 2021

Dear Prof. Roman,

we are pleased about the positive views of the reviewers and appreciate also the careful editing of the manuscript by the reviewer #1: 

We accepted all but one suggestions:

Lines 153-162 — paragraph should be in past tense

Corrected

Line 219 tense shift

Corrected

Line 244 “according to based on” (redundant)

We have not changed this sentence, because "based on first trials" is here a complementary information, a fact, which should be pointed out

Line 326 “more frequently" need to add "chosen”

Corrected

In the figure caption for Fig 1 please give a sense of what these numbers mean. - ie - left and + right? We don't need the full explanation that comes later in the text, but a sense of what these numbers are telling us would be helpful at this point.

The reviewer means probably Table 1 and/or Figure 3. We complemented the caption in both cases as suggested.

I still find the figures confusing, but I’m not an expert on how to present these results with turning preference so I leave it to the other reviewer and the editor's expertise.

We assure the reviewer that this way of illustrating the results is common in the literature dealing with spatial orientation.

Once again, many thanks for considering our manuscript and ist academic processing.

With best regards and wishes for a prosperous and healthy new year

Hynek Burda

On behalf of all coauthors.

---

## [Editor Report · Decision Letter 2]

11 Jan 2021

Turning preference in dogs: north attracts while south repels

PONE-D-20-31110R2

Dear Dr. Burda,

We’re pleased to inform you that your manuscript has been judged scientifically suitable for publication and will be formally accepted for publication once it meets all outstanding technical requirements.

Kind regards,

Gregg Roman, PhD

Academic Editor

PLOS ONE
---

## [Editor Report · Acceptance letter]

20 Jan 2021

PONE-D-20-31110R2 

Turning preference in dogs: north attracts while south repels 

Dear Dr. Burda:

I'm pleased to inform you that your manuscript has been deemed suitable for publication in PLOS ONE. Congratulations! Your manuscript is now with our production department. 

Kind regards, 

on behalf of

Dr Gregg Roman 

Academic Editor

PLOS ONE